# Volatile Metabolome and Aroma Differences of Six Cultivars of *Prunus mume* Blossoms

**DOI:** 10.3390/plants12020308

**Published:** 2023-01-09

**Authors:** Ting Li, Xi Zhao, Xueli Cao

**Affiliations:** Beijing Advanced Innovation Center for Food Nutrition and Human Health, Beijing Technology and Business University, Beijing 100048, China

**Keywords:** aroma, HS-SPME-GC-MS, *Prunus mume* blossom, untargeted metabolomics, volatile metabolome

## Abstract

*Prunus mume* is a traditional Chinese plant with high ornamental and application values due to its very early blooming and unique fragrance. Long-term breeding and cultivation have resulted in a variety of *P. mume* blossoms and have made their exploitation more possible. Existing studies on the volatile metabolome and aroma of *P. mume* blossoms are limited. In this study, six extensively planted cultivars of *P. mume* blossoms, including Gulihong (GLH), Yudie (YD), LvE (LE), Dongfang Zhusha (DFZS), Jiangmei (JM), and Gongfen (GF), were investigated for their differences in terms of volatile metabolome, as well as their aroma characteristics based on the strategies and methods of metabolomics. The volatile metabolites were analyzed using HS-SPME-GC-MS technique. A total of eighty-nine compounds were detected and sixty-five of them were tentatively identified, including thirty-seven phenylpropanoids/benzenes, seventeen fatty acid derivatives, ten terpenoids, and one other compound. YD contains the most volatile metabolites in terms of number and amounts, which impart more abundant aromas to this cultivar. Fifteen differential compounds were screened through the untargeted metabolic analysis of twenty-nine samples by principal component analysis (PCA) and partial least squares-discriminant analysis (PLS-DA), while nine compounds were screened based on the odor activity value (OAV) analysis of the sixty-five identified compounds. GLH and GF, JM and LE were found to be more similar to each other based on chemometrics analysis of both volatile contents and OAVs, while YD and DFZS were markedly different from other cultivars. Six main metabolites, including benzaldehyde, methyl benzoate, benzyl acetate, eugenol, (E)-cinnamic alcohol, and 4-allylphenol, together with 2-nonenal, 3,4-dimethoxytoluene, and trans-β-Ionone were screened as differential compounds, owing to their higher contents and/or lower olfactory threshold, which endow an almond, cherry, phenolic, wintergreen, cananga odorata, floral, jasmine, hyacinth, cinnamon, clove, woody, medicinal, and violet fragrance to each variety, and greatly contribute to the aroma differences of six cultivars of *P. mume* blossom.

## 1. Introduction

*Prunus mume*, belonging to Prunus in the family Rosaceae, is well known and native to central and southern China, Taiwan, and other places, and has been cultivated by the Chinese people since ancient times, mainly as an ornamental plant [1]. *P. mume* blooms much earlier than other flowers (mostly in February to March in central and east China), and is appreciated for its cold resistance, unique floral scent, and genetic diversity of cultivars. The blossom of *P. mume* has distinctive and refreshing floral scents compared with other Prunus species. However, long-term cultivation and domestication have resulted in a variety of *P. mume* blossoms with different branch postures, flower colors, number of petals, and floral scent [2].

Floral scent plays an important role in the reproductive processes of many plants and enhances the aesthetic properties of ornamental plants, and is closely related to the genes and enzymes involved in the biosynthesis of floral scent volatiles. Most floral substances belong to one of three major groups: phenylpropanoids/benzenoids, terpenoids, and fatty acid derivatives, which originated in the shikimate pathway, the mevalonate/non-mevalonate pathway, and the malonate pathway in the plant metabolism, respectively [3].

Some researchers have studied the scent components of some *P. mume* blossom cultivars from a better breeding point of view [4,5,6,7]. Zhao et al. [4] analyzed the volatile constituents of five different cultivars of *P. mume* blossoms and 45 compounds were tentatively identified; among them benzaldehyde, benzyl alcohol, and benzyl acetate originated from the shikimate pathway and were the main volatiles of the analyzed *P. mume*, while the common components of hexyl acetate, eugenol, benzyl acetate, and α-pinene were considered the reason for the similar aroma of *P. mume* blossoms. Hao [5] analyzed the emitted and endogenous floral scent compounds of *Prunus mume* and hybrids, and revealed that benzyl acetate had a stronger tendency to be volatile than the other compounds, and the volatilization rate of volatile metabolites greatly varied among different cultivars. Zhang et al. [6] explored the diversity of floral scents in *P. mume* intraspecific cultivars with three different corolla colors. A total of 31 volatile metabolites were identified, in which phenylpropanoids/benzenoids accounted for over 95% of the total amounts, and the emissions of benzyl alcohol, cinnamyl alcohol, benzyl acetate, eugenol, (E)-cinnamyl acetate, and benzyl benzoate could make these intraspecific cultivars distinguishable from each other. Wang et al. [7] investigated the headspace volatiles and endogenous extracts of different aroma types of *P. mume* blossoms and 66 headspace volatiles and 74 compounds in endogenous extracts were putatively identified, of which phenylpropanoids/benzenoids were the main volatile organic compound categories. Several biomarkers including (Z)-2-hexen-1-ol, amyl acetate, cinnamaldehyde, methyl salicylate, cinnamyl alcohol, and benzoyl cyanide were found to contribute to the differences of strong-, fresh-, sweet-, and light-scented types of *P. mume* blossoms.

As a traditional ornamental plant with a unique fragrance and colorful colors, studies on *P. mume* are generally limited. Previous studies on blossoms have been aimed at the breeding of more diverse cultivars and have paid more attention to headspace volatiles. Today, the successful cultivation of *P. mume* on a large scale makes extensive exploration more applicable. Like many flower plants, floral metabolites of *P. mume* blossoms could be used in perfumes, cosmetics, foods, and pharmaceuticals [8]; meanwhile, further investigation remains necessary for the applications of specific *P. mume* blossoms due to the diversity of cultivars.

In this paper, six cultivars of *P. mume*, extensively planted in a base garden located in east China, were investigated for their differences in blossom volatile metabolome and their aroma characteristics based on the strategies and methods of metabolomics, combined with analysis of the odor activities, aiming at potential exploitations.

## 2. Results and Discussion

### 2.1. Analysis of P. mume Blossom Volatile Metabolites by HS-SPME-GC-MS

Deferent sample pretreatment methods have been employed in previous studies; both static headspace adsorption and head space-solid phase microextraction (HS-SPME) were used to collect headspace volatiles emitted from whole blooming flowers [4,6,7,9], while organic solvent extraction of the ground powder of flowers in liquid nitrogen was applied for endogenous component sampling [5,7]. HS-SPME was selected to collect volatile constituents from the powder of flowers ground in liquid nitrogen for the easy addition of internal standards and preparation of quality control samples. The same strategy was also used in the analysis of volatile metabolites in dark tea during the fermentation process [10]. Extraction efficiency was evaluated by comparing the compound numbers and the total peak areas between whole blooming flowers and the ground powder of flowers in the same amounts, as well as under the same analytical conditions as HS-SPME-GC-MS. The RSD were 2.7% and 2.5% for their compound numbers and the total peak areas, respectively. Except for the slight variation in the relative peak area of certain constituents, no significant difference between the two sampling methods was observed. Meanwhile, the stability of the sample during short storage (within 6 days) at −40 °C before analysis was examined. The RSD of the areas of 15 main components were in the range of 2.9–17.5%, the same level as the HS-SPME-GC-MS analytical method. Thus, the sampling method adopted in this work is reliable and feasible.

The TICs of volatile metabolites in *P. mume* blossoms are shown in Appendix A. All TIC were processed by Agilent MassHunter Unknowns Analysis software and peaks were manually aligned. A total of 89 compounds were detected in all *P. mume* blossom samples (Appendix A), and 65 of them were identified by varying degrees by comparing their mass spectra with NIST 11 and their RIs with NIST Chemistry WebBook (https://webbook.nist.gov/chemistry/) (accessed on 6 April 2022) [11]; 24 unidentified compounds were labeled as unknown. On this basis, some significant compounds were also compared with the standards. The sixty-five identified compounds are shown in Table 1, including thirty-seven phenylpropanoids/benzenes, seventeen fatty acid derivatives, ten terpenoids, and one other compound.

Based on the concentration of the internal standard, a quantitative analysis of all detected compounds was carried out according to the following Formula (1), in which *w* is the content, *A* is the peak area, *i* is the certain component to be measured, *IS* is the internal standard; for 1,2-dichlorobenzene, here, the relative correction factor *f*′ was set to 1:(1)$$w_{i,s}=f^{\prime }\times \frac{A_{i,s}}{A_{IS}}\times w_{IS}$$

The quantitation results are also shown in Appendix A. The number of compounds detected in the YD, JM, GLH, LE, DFZS, and GF samples were 82, 68, 72, 73, 73, and 74, and the total amounts were 607.94, 481.89, 541.37, 520.47, 480.34, 562.90 μg/g, respectively. The contents of the identified compounds accounted for 96.1–98.6% of the total amount. The number and total amount of compounds contained in YD were the highest, while the number of compounds in JM was the least, and the total amount of compounds in DFZS was the lowest, which is consistent with the fact that JM and DFZS cultivars had a lighter aroma on the senses. Fifty-seven common compounds were found among the six cultivars. Contents of benzaldehyde, methyl benzoate, benzyl benzoate, eugenol, 4-allylphenol, and benzyl alcohol in the six cultivars of *P. mume* blossoms were all far more than in other constituents (the relative content was greater than 1% in all six species), which are the main volatile metabolites of *P. mume* blossoms and all belong to phenylpropanoids/benzene compounds. This result is partially similar to existing studies [4,5,6,7] on the volatile metabolites in *P. mume* blossoms, with a difference in the content of some compounds; the most abundant substance in all six species was methyl benzoate in the present study, other than benzyl acetate, which was closely related to the difference of most of studied cultivars. Comparison of the results of the same cultivar, JM, in the present and a previous study [7], revealed some differences between both compounds and their contents. This may be partially attributed to the origin difference of JM, from Zhejiang, while the previous was from Hubei province, China. Meanwhile, different pretreatment methods, post-ground samples, and flowers were respectively applied, and extractions with different temperatures and time by SPME were performed. The analytical result is closely related to the sample pretreatment, extraction, and analytical conditions. Nevertheless, these would not affect the analysis of volatile metabolome and aroma differences among different cultivars focused on in this work.

### 2.2. Analysis of Volatile Metabolite Differences in Six Cultivars of P. mume Blossoms

In order to understand the differences in the composition of volatile metabolites in *P. mume* blossoms among cultivars, unsupervised principal component analysis (PCA) was used to perform pattern recognition on all *P. mume* blossom samples. A total of fifty-seven data from twenty-nine samples (including two repetitions of each sample; one GLH sample data was excluded) and nine data from the QC sample were imported into the SIMCA software and, after performing Pareto scaling, a principal component analysis model was established. In the two-dimensional score plot of the first two principal components (Figure 1), the variance explained by the first two principal components was 47.7% and 25.7%, indicating that the PCA model had a good explanation rate for the difference in the original data. The QC samples were close to the origin and clustered well, indicating that the HS-SPME-GC-MS analysis method used in this study was stable and reliable. Samples were clustered mainly according to cultivars in the PCA score plot. The cultivars YD and DFZS clustered well and separated from other cultivars, indicating significant inter-cultivar differences. The cultivars GLH and GF, LE and JM were closely clustered in the first principal component and not significantly separated from each other in the second principal component, indicating some similarity between GLH and GF, LE and JM in terms of the volatile metabolites in blossoms.

A supervised PLS-DA was used to investigate the differential metabolites among the six cultivars of *P. mume* blossoms. The 57 data were divided into categories according to the sample information in advance, and a supervised PLS-DA model was established (R^2^X = 0.961, R^2^Y = 0.957, Q^2^ = 0.867), as shown in Figure 2A. R^2^X and R^2^Y higher than 0.9, in this model, represent that the model has a high interpretation rate for the X and Y variables, and the high Q^2^ represents a good model fit and high predictive ability. The cross-validation results of 100 permutation tests indicated that this PLS-DA model was reliable without overfitting (intercepts of R^2^ and Q^2^ were 0.229 and −0.741, respectively; Q^2^ and R^2^ of the permutation test models were both lower than the current model) (Figure 2B). The variable importance of projection (VIP) value was used to find compounds that had a greater contribution to differences. As shown in the loading plot of the PLS-DA model in Figure 2C, 15 compounds with VIP > 1 were selected as differential compounds, as listed in Table 2, including benzaldehyde, 4-allylphenol, methyl benzoate, benzyl benzoate, eugenol, (E)-cinnamyl acetate, benzyl alcohol, (E)-cinnamic alcohol, (E)-cinnamaldehyde, benzyl acetate, unkown19, alkane3, methyleugenol, (E)-methyl isoeugenol, and ethyl benzoate. These compounds made greater contributions to the differences in volatile metabolites of different cultivars of *P. mume* blossom. Except for the unidentified unknown19 and Alkane3, the remaining 13 compounds all belong to phenylpropanoid/benzene compounds, which are mainly synthesized by the shikimate pathway.

To further investigate how these fifteen compounds differentiate among the six *P. mume* blossoms, Hierarchical Cluster Analysis (HCA) was simultaneously performed on these fifteen compounds and samples, as shown in Figure 3. From the similarity among samples, GLH was more similar to GF, and LE was similar to JM, which is consistent with the results of PCA and PLS-DA analysis. The contents of three compounds of (E)-cinnamaldehyde, (E)-cinnamic alcohol, and (E)-cinnamyl acetate were much higher in GLH and GF cultivars than in the other four cultivars, making GLH and GF similar to each other and dissimilar from others. A previous study [6] also found that (E)-cinnamic alcohol and (E)-cinnamyl acetate were only synthesized in pink flowers. However, in this study, these three compounds were also found in other cultivars of *P. mume* blossoms, with very low content. According to existing research on flowers and fruits [12,13], (E)-cinnamic acid is converted to cinnamoyl-CoA by (E)-cinnamic acid: CoA ligase in flowers and fruits, and the latter is catalyzed by cinnamoyl CoA reductase (CCR) to generate cinnamaldehyde; after a series of reactions, other phenylpropene compounds including (E)-cinnamyl acetate are generated. Higher concentrations of (E)-cinnamaldehyde, (E)-cinnamic alcohol, and (E)-cinnamyl acetate indicated that, compared with the other four cultivars, GLH and GF *P. mume* are likely to have differences in related metabolic regulation.

Similarly, benzaldehyde, benzyl alcohol, and benzyl acetate, among these 15 compounds, belong to the same metabolic branch in plant metabolism [14,15]. When HCA was performed with compounds, benzyl alcohol, benzyl acetate, and benzaldehyde were also closer together, indicating the similarity of the variation among samples for these three compounds. The contents of benzaldehyde, benzyl alcohol, and benzyl acetate were higher in YD and lower in DFZS. These results suggested that there may also be differences in this metabolic branch among *P. mume* cultivars. Benzyl alcohol acetyltransferase genes in *P. mume* (PmBEAT) have been identified, and overexpression or inhibition of this gene has been demonstrated to promote or inhibit the synthesis of benzyl acetate [16]. Differences in the contents of benzaldehyde, benzyl alcohol, and phenylmethyl acetate in *P. mume* cultivars may reveal differences in the expression of the PmBEAT gene and other related genes in the six cultivars of *P. mume* blossoms.

Eugenol and 4-allylphenol also showed consistency of variation across cultivars, with eugenol and 4-allylphenol being higher in YD and DFZS, and methyleugenol being higher in YD and GLH. In the majority of plants, these compounds belong to the same p-coumaric acid branch of the shikimic acid pathway [17]. p-Coumaric acid is gradually reduced to p-coumaryl alcohol, which is then catalyzed by 4-Coumaryl/Coniferyl alcohol acetyl transferase (CAAT) and Eugenol synthase (EGS) to produce 4-allylphenol. Methylation of 4-allylphenol by chavicol O-methyl transferase (CVOMT) can also produce 4-allylanisole, which was also detected in all the samples. As for the synthesis of eugenol, p-Coumaric acid is first converted to Caffeoyl CoA, which is catalyzed by Caffeoyl-CoAO-methyl transferase (CcoAOMT), Cinnamoyl-CoA/coniferyl-CoA reductase (CCR), Coumaryl/coniferyl alcohol dehydrogenase (CAD), and coniferyl alcohol acyltransferase (CFAT) in turn to produce Coniferyl acetate, and is then catalyzed by EGS or Isoeugenol synthase (IGS) to produce eugenol or isoeugenol. The latter can also be methylated to obtain the corresponding methyleugenol and methyl isoeugenol. The differences in the contents of eugenol, 4-allylphenol, and methyleugenol in various cultivars of *P. mume* may imply differences in the regulation of expression of enzymes catalyzing the relevant synthetic reactions, which need to be further investigated.

### 2.3. Analysis of Aroma Differences of Six Cultivars of P. mume Blossoms

According to the previous analysis, contents of methyl benzoate, benzaldehyde, benzyl benzoate, eugenol, 4-allylphenol, unknown19, and benzyl alcohol were found to be high in all six *P. mume* species, and these seven main compounds in *P. mume* also belonged to the fifteen differential compounds that were screened. This might indicate that these major components were also responsible for the aroma of different plum blossoms. However, due to the different olfactory thresholds of the various compounds, it is the odor activity values (OAVs) of the compounds, rather than their contents that actually determine the contribution to the aroma. Based on the above analysis of the differences in volatile metabolites in different cultivars of *P. mume* blossoms, OAV and aroma descriptions were further investigated for differences in the aroma characteristics of these six *P. mume* species. The OAV is defined as the ratio of the actual concentration of a compound in the sample to the olfactory threshold of the compound, reflecting the actual odor intensity of the compound in the sample [18], and is widely used for aroma analysis of various fruits [18,19], flowers [7], and foods [20,21,22]. The OAVs of sixty-five identified volatile metabolites in six cultivars of *P. mume* blossoms were listed in Appendix A. A Supervised PLS-DA was performed based on the OAV data, as presented in Figure 4 (R^2^X = 0.998, R^2^Y = 0.922, Q^2^ = 0.861). It was found that the OAV data of each cultivar blossom were not clustered, as well as that of the contents of volatile compounds (Figure 4A). This is probably due to the absence of the OAV of some unknown compounds, leading to insufficient information. Nevertheless, it can still be similarly observed that the OAV data of JM and LE, GLH and GF are respectively closer to each other, while that of YD and DFZS are somewhat further away from others. The cross-validation results of 100 permutation tests indicated that this PLS-DA model was reliable without overfitting (intercepts of R^2^ and Q^2^ were 0.145 and −0.462, respectively; Q^2^ and R^2^ of the permutation tests models were both lower than the current model) (Figure 4B). Based on this PLS-DA analysis of OAVs, nine aroma differential volatile metabolites with VIP > 1 were screened in six cultivars of *P. mume* blossoms, as seen in Figure 4C, and their aroma details are listed in Table 3 [23,24], including benzaldehyde, 2-nonenal, methyl benzoate, benzyl acetate, 3,4-dimethoxytoluene, trans-β-Ionone, eugenol, (E)-cinnamic alcohol, and 4-allylphenol. These compounds greatly contribute to the aroma differences of the six cultivars of *P. mume* blossom. Six of them are in the list of fifteen differential compounds based on contents, except for 2-nonenal, dimethoxytoluene, and trans-β-ionone. It is noteworthy that trans-β-ionone, a compound with an extremely low Olfactory Threshold of 0.000007 μg/g, is screened out with the second highest OAVs, just lower than that of methyl benzoate. Trans-β-Ionone is a natural plant volatile compound, and it is the 9,10 and 9′,10′ cleavage product of β-carotene by the carotenoid cleavage dioxygenase [25]. The higher Trans-β-Ionone OAV in YD and JM gives these two cultivars a more intense violet and raspberry flavor, which may be weaker in DFZS.

HCA was simultaneously performed on the nine compounds and samples, as shown in Figure 5, to further understand how these nine compounds make different aromas among the six *P. mume* blossoms. Similar to the HCA results in Section 2.2, the same sample clustering results are shown in Figure 5, where GLH is more similar to GF and JM is more similar to LE. GLH and GF are characterized by higher OAV of (E)-cinnamic alcohol, together with benzaldehyde in GF and 2-nonenal in GLH, respectively. JM and LE are characterized by higher OAVs of methyl benzoate and benzyl acetate, together with trans-β-ionone in JM and 3,4-dimethoxytoluene in LE, respectively. Comparatively, YD and DFZS are significantly different from others. YD is characterized by higher OAVs of eugenol, 4-allylphenol, trans-β-ionone, benzyl acetate, benzaldehyde, and 3,4-dimethoxytoluene, while DFZS is characterized by higher OAVs of eugenol, 4-allylphenol, and methyl benzoate.

### 2.4. Correlation between the Differential Compounds and Aroma Characteristics

Table 4 compares the differential compounds screened based on the contents of 89 detected compounds and OAVs of 65 identified compounds in different cultivars. Six compounds including benzaldehyde, methyl benzoate, benzyl acetate, eugenol, (E)-cinnamic alcohol, and 4-allylphenol are in the lists for fifteen and nine differential compounds based on contents and OAV, respectively. All of these compounds originated from the main metabolic pathways and were of higher contents and/or lower olfactory threshold, which greatly contribute to the aroma differences of the six cultivars of *P. mume* blossom. Furthermore, trans-β-ionone was screened in the OAV list owing to its extremely low olfactory threshold. Apart from three compounds, for which olfactory thresholds are not available, most of the remaining six compounds in the 15 differential compound list based on contents are of relatively higher olfactory thresholds, and are out of the list of OAV.

Specifically, taking both differential compounds based on contents and OAV into account, it can be observed that (E)-cinnamic alcohol is the common characteristic compound in GLH and GF, which may confer stronger hyacinth and cinnamon scent in these two cultivars; while methyleugenol may confer slightly stronger clove and cinnamon aroma in GLH and benzaldehyde may give GF a more almond and cherry fragrance. Methyl benzoate may bring a common strong phenolic, wintergreen, and almond scent, and benzyl acetate may bring floral and jasmine base scents for JM and LE, while trans-β-Ionone may give a strong violet and raspberry scent for JM, and 3,4-dimethoxytoluene a stale and musty scent for LE. DFZS is unique for the same characteristic compounds on both content and OAV sides, including methyl benzoate, eugenol, and 4-allylphenol, which impart DFZS with more prominent phenolic, wintergreen, and almond scents, clove and woody scents, and phenolic and medicinal scents. YD is quite different from other *P. mumes* for its very abundant characteristic compounds, including benzaldehyde, benzyl acetate, eugenol, and 4-allylphenol on the basis of content and OAV, which generate rich fragrance for YD together with 2-nonenal, 3,4-dimethoxytoluene, and trans-β-ionone. This is consistent with the results derived above, that YD contains the most volatile metabolites in terms of both number and amount.

*P. mume* blossom is rich in germplasm resources and has various unique aromas, with strong ornamental and economic value [1,8]. Research on the volatile metabolome and aroma characteristics of different cultivars of *P. mume* blossoms is helpful for the development and utilization of *P. mume* blossom resources. However, the formation of human olfaction and aroma is complex; based on the clarification of volatile content, the formation of aroma is not simply a simple sum of all aroma components. The same structure or homologs are prone to synergistic and additive effects, and different aroma rhythms or aroma components with large structural differences are prone to masking and inhibiting effects [27]. Therefore, the aroma characteristics of *P. mume* blossoms still need to be studied by applying more in-depth aroma formation theories and methods.

## 3. Materials and Methods

### 3.1. Plant Materials

Six different cultivars of *P. mume* blossoms (Figure 6), Gulihong (GLH), Yudie (YD), LvE (LE), Dongfang Zhusha (DFZS), Jiangmei (JM), and Gongfen (GF), were collected with branches in March 2022 from a *P. mume* planting base, Zhejiang Huzhou, China. The branches were collected from 4–5 trees for each *P. mume* cultivar for biological replication (29 samples in total). The branches were express delivered to the laboratory under 4–6 °C and the *P. mume* blossoms (whole flowers, including calyx) were detached from their branches and immediately ground in liquid nitrogen until pulverized (within about 1 min); the sample powder was then transferred to PE centrifuge tubes and sealed. Equal amounts of blossom samples powder were evenly mixed as the quality control (QC) sample. All samples were kept in the −40 °C freezer before analysis.

### 3.2. Chemicals

All authentic standards with at least 95% purity, including 2-methyl-3-heptanone, 1,2-dichlorobenzene, benzaldehyde, methyl benzoate, ethyl benzoate, benzyl alcohol, methyleugenol, (E)-cinnamaldehyde, and (E)-cinnamic alcohol, were from Aladdin Co., Ltd. (Shanghai, China); (E)-2-hexenal, 4-allylphenol, benzyl acetate, (E)-cinnamyl acetate, eugenol, and benzyl benzoate were from MACKLIN Co., Ltd. (Shanghai, China). Methanol used to dilute standards was purchased from MREDA Scientific Co., Ltd. (Beijing, China). A mixture of C7–C40 *n*-alkane standard solution was from o2si (North Charleston, SC, USA).

### 3.3. HS-SPME-GC-MS Analysis Method

An amount of 0.300 g (±0.002 g) of the sample powder was weighed into a 15 mL SPME bottle (CNW Technologies GmbH, North Rhine-Westphalia, Germany), 5 μL of 0.816 μg/μL 2-methyl-3-heptanone solution (in methanol) was added with 10 μL of 0.521 μg/μL 1,2-dichlorobenzene solution (in methanol) as internal standards to correct the retention time and conduct quantitative analysis of the volatile metabolites. The bottle cap was tightened with a PTFE/silica gel septum (JIEDAO Technology Instrument Co., Ltd., Nanjing, China) and placed on an SPME Sampling Stand (PC-420D, Corning Inc., Corning, NY, USA) to equilibrate at 40 °C for 20 min. A SPME fiber (50/30 μm, DVB/CAR/PDMS, SUPELCO, Bellefonte, PA, USA) was inserted to extract for 60 min at the same temperature, and then the fiber was immediately inserted into the GC inlet for thermal desorption.

The volatile metabolites were analyzed by GC-MS (7890B-7000C, Agilent Technologies Inc., Palo Alto, CA, USA) equipped with an HP-INNOWax column (30 m × 250 μm × 0.5 μm; Agilent Technologies Inc., Palo Alto, CA, USA). The inlet temperature was set to 250 °C and spitless injection was adopted. High-purity helium with a flow rate of 1 mL/min was the carrier gas. The initial oven temperature was held at 40 °C for 5 min, and was then programed to 230 °C at 3 °C/min and maintained for 10 min; the total time was 78.33 min. The mass spectrometer adopted an electron ionization source with 70 eV electron energy, and the temperature of the transfer line and the ion source were both 250 °C. The scanning range was 30–450 *m*/*z* conducted on the first quadrupole with a scan time of 600 ms.

The HS-SPME-GC-MS conditions employed here were optimized in advance. Each sample was sampled twice in duplicate and the QC samples were injected every 9 intervals to assess the stability of the analysis system. A blank control sample in which only internal standards were included was analyzed for signal verification. Additionally, the mixed standard solution of C7–C40 *n*-alkanes and the mixed authentic standards solution were analyzed at the same conditions for calculation of the retention index (RIs) and assisting qualitative analysis of volatile metabolites in samples.

### 3.4. Data Processing and Chemometric Analysis

The total ion chromatograms (TIC) obtained in GC-MS analysis were checked using Agilent MassHunter Qualitative Analysis software (B.07.00, Agilent Technologies Inc., Palo Alto, CA, USA). Deconvolution processing, peak picking, and mass spectrometry comparison were performed by Agilent MassHunter Unknowns Analysis software (B.07.01, Agilent Technologies Inc., Palo Alto, CA, USA) and NIST 11 library (match factor > 75). Deducting blank control peaks, and aligning the peaks of the obtained 66 spectra (57 for samples, 9 for QC), the peak area matrix of the volatile metabolites was obtained. The matrix was imported into SIMCA software (14.1.0, Umetrics, Malmo, Sweden) and MetaboAnalyst 5.0 (https://www.metaboanalyst.ca/faces/home.xhtml) (accessed on 6 April 2022) [28] for principal component analysis (PCA), partial least squares-discriminant analysis (PLS-DA), and hierarchical clustering analysis (HCA). Unless otherwise indicated, the olfactory thresholds required to calculate OAV were from L.J. van Gemert’s book: *Compilations of Flavor Threshold Values in Water and Other Media* (second enlarged and revised edition) [23], and the aroma descriptions were from the Perflavory Information System (www.perflavory.com) (accessed on 6 April 2022) [26].

## 4. Conclusions

In this paper, volatile metabolites and their OAVs in six cultivars of *P. mume* blossoms were analyzed through strategies and methods of metabolomics and chemometrics. A total of eighty-nine compounds were detected in all *P. mume* blossom samples, and sixty-five of them were tentatively identified, including thirty-six phenylpropanoids/benzenes, seventeen fatty acid derivatives, ten terpenoids, and one other compound. YD contains the most volatile metabolites in terms of number and amount. GLH and GF, JM and LE were found to be more similar to each other based on the chemometrics analysis of both contents and OAVs, while YD and DFZS were markedly different from other cultivars. The main metabolites, including benzaldehyde, methyl benzoate, eugenol, and 4-allylphenol, together with benzyl acetate, (E)-cinnamic alcohol, 2-nonenal, 3,4-dimethoxytoluene, and trans-β-Ionone, were screened as the main differential compounds owing to their higher contents and/or lower olfactory threshold, which greatly contribute to the aroma differences of six cultivars of *P. mume* blossom.

## Figures and Tables

**Figure 1 plants-12-00308-f001:**
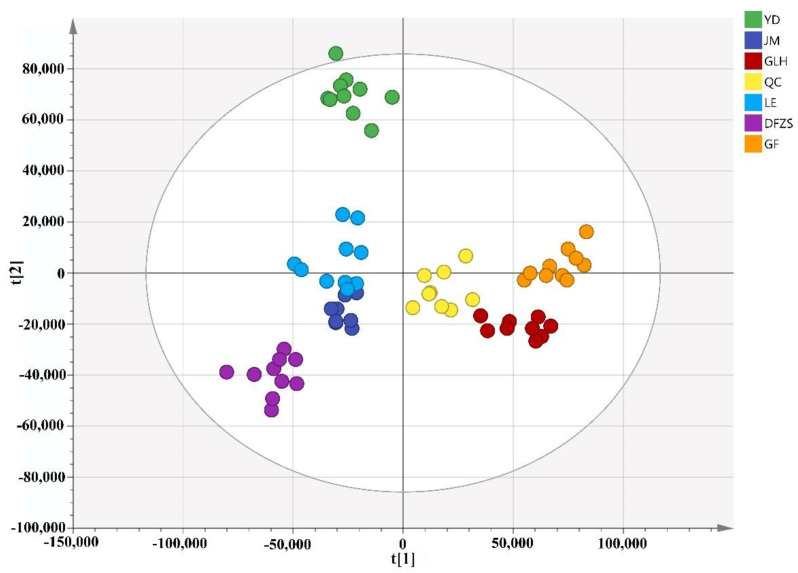
PCA score plot of volatile compounds of six varieties of *P. mume* blossom samples.

**Figure 2 plants-12-00308-f002:**
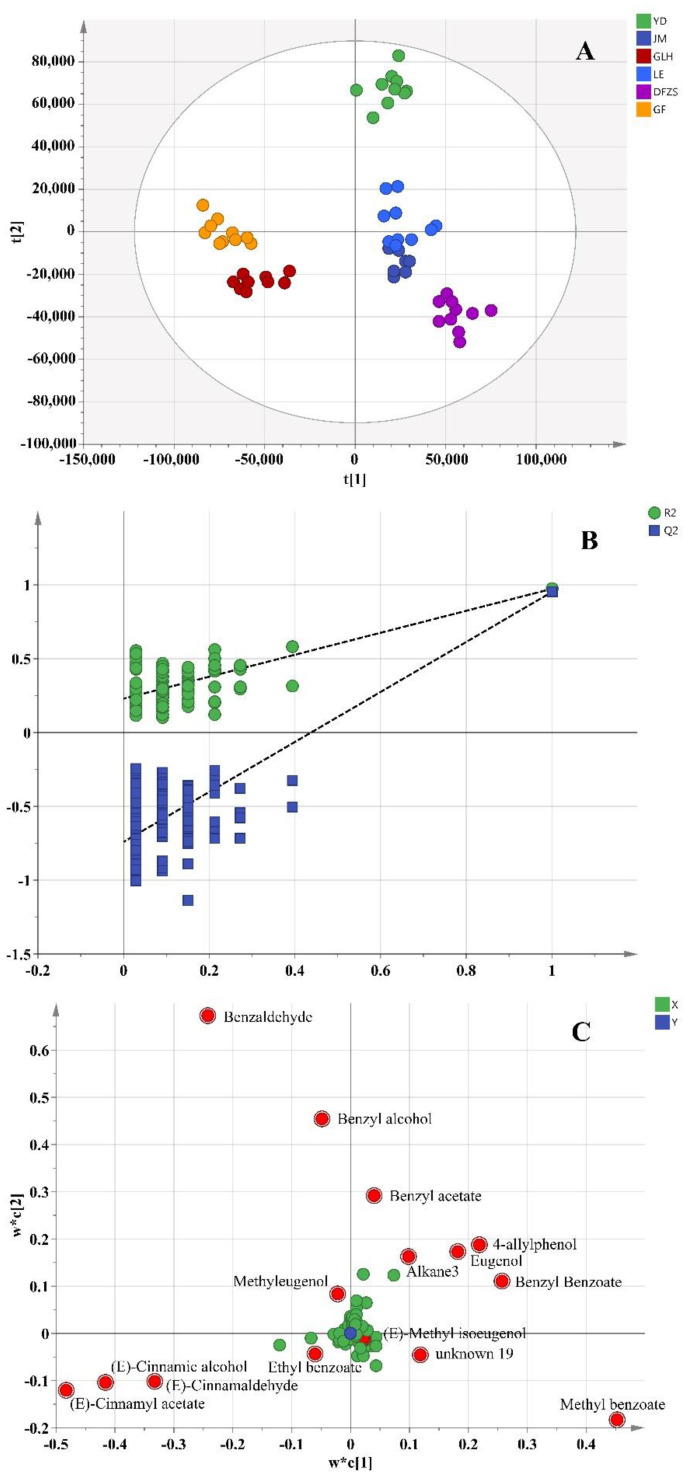
PLS-DA analysis of volatile compounds of six varieties of *P. mume* blossom samples. (**A**) PLS-DA score plot; (**B**) cross-validation plot of the PLS-DA model with 100 permutation tests; (**C**) loading plot of PLS-DA (red dot represents the most differential compounds, VIP > 1).

**Figure 3 plants-12-00308-f003:**
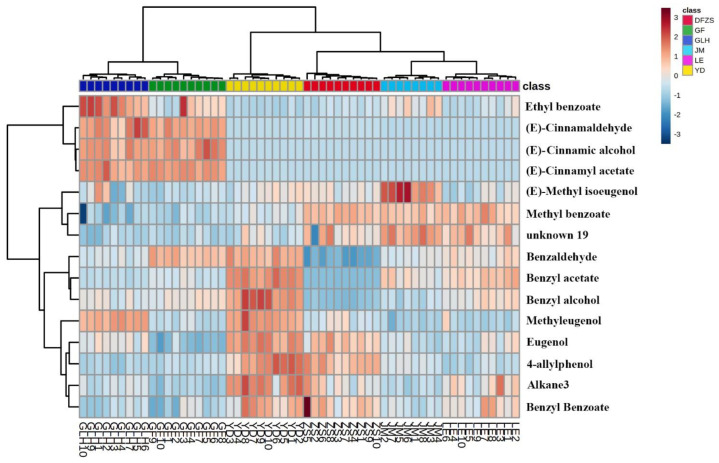
HCA analyses of fifteen differential volatile compounds of six varieties of *P. mume* blossom samples.

**Figure 4 plants-12-00308-f004:**
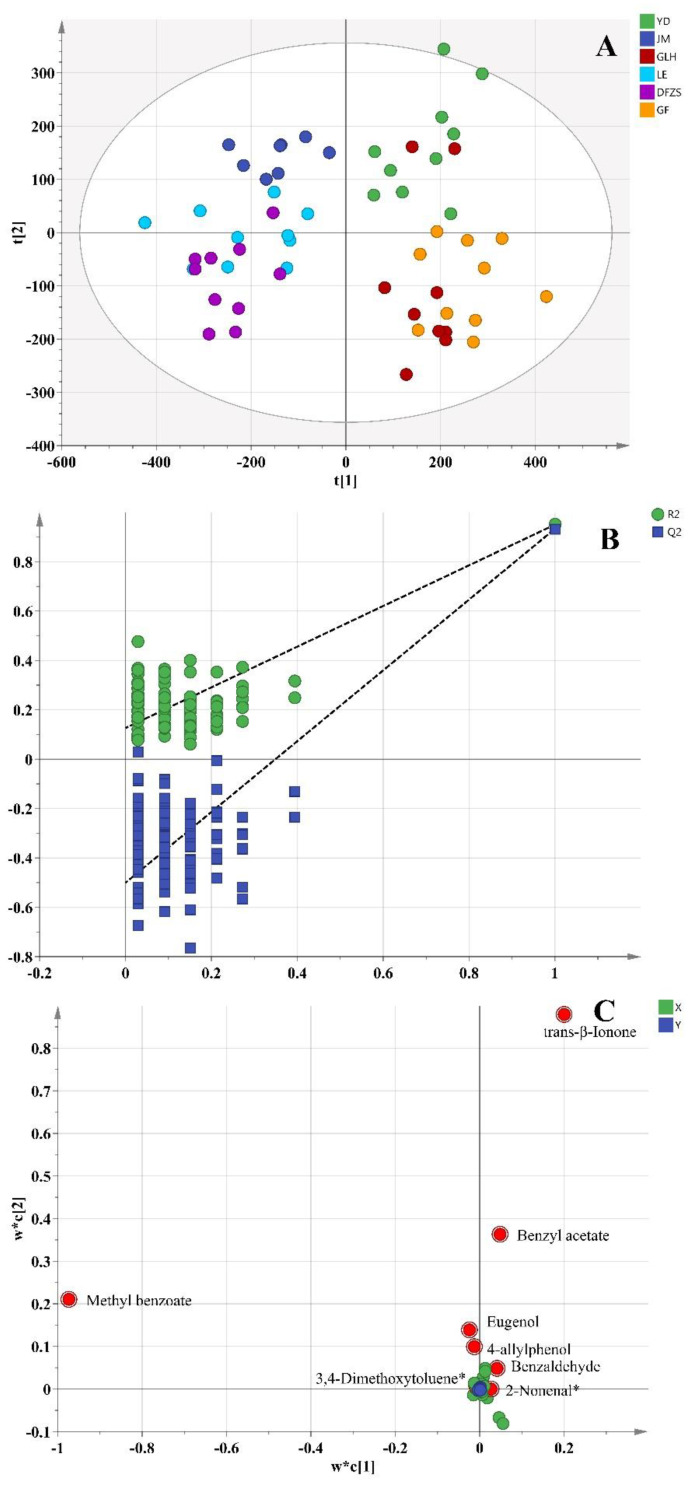
PLS-DA analysis of volatile compounds based on their OAVs in six varieties of *P. mume* blossom samples. (**A**) PLS-DA score plot; (**B**) cross-validation plot of the PLS-DA model with 100 permutation tests; (**C**) loading plot of PLS-DA (red dot represents the most differential compounds, VIP > 1). “*” indicates that the compound may be other isomers.

**Figure 5 plants-12-00308-f005:**
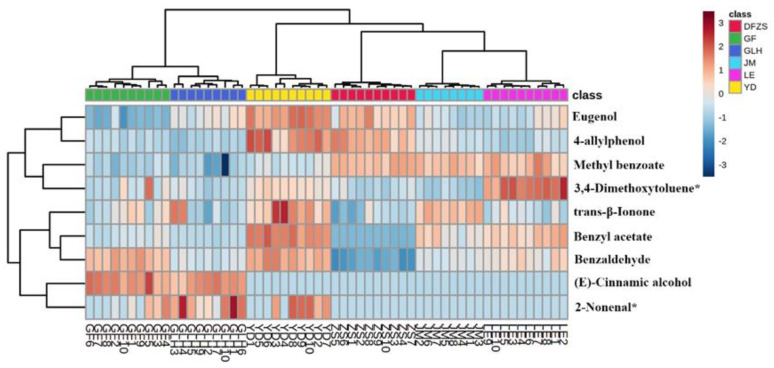
HCA analyses of nine differential aroma compounds of six varieties of *P. mume* blossom samples. “*” indicates that the compound may be other isomers.

**Figure 6 plants-12-00308-f006:**
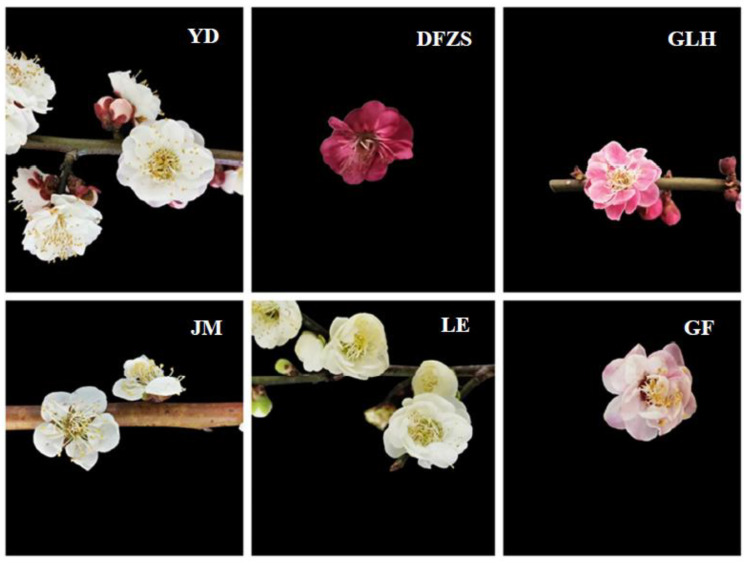
Pictures of six varieties of *P. mume* blossoms. YD (Yudie), DFZS (Dongfang Zhusha), GLH (Gulihong), JM (Jiangmei), LE (lve), GF (Gongfen).

**Table 1 plants-12-00308-t001:** Identification of volatile compounds in *P. mume* blossoms.

No.	Compound Name	CAS Number	Measured RI **	Reference RI	Identification
	**Phenylpropanols/Benzenes**				
1	o-Xylene	108-38-3	1158.18 ± 0.59	1158	MS, RI
2	Styrene	100-42-5	1269.78 ± 0.32	1263	MS, RI
3	Benzaldehyde	100-52-7	1536.47 ± 1.03	1529	MS, RI, Std
4	Methyl benzoate	93-58-3	1649.52 ± 1.06	1641	MS, RI, Std
5	Ethyl benzoate	93-89-0	1691.64 ± 0.61	1681	MS, RI, Std
6	Benzyl formate	104-57-4	1700.61 ± 1.34	1705	MS, RI
7	4-Allylanisole	140-67-0	1700.61 ± 1.34	1687	MS, RI
8	Benzyl acetate	140-11-4	1747.71 ± 0.45	1746	MS, RI, Std
9	Methyl salicylate	119-36-8	1802.31 ± 0.52	1798	MS, RI
10	3,4-Dimethoxytoluene *	494-99-5	1828.76 ± 0.42	-	MS
11	Benzyl alcohol	100-51-6	1886.60 ± 0.78	1878	MS, RI, Std
12	Butyl benzoate	136-60-7	1901.17 ± 1.62	1882	MS, RI
13	(Z)-Cinnamaldehyde	57194-69-1	1912.02 ± 0.42	1884	MS
14	Phenylethyl Alcohol	60-12-8	1915.72 ± 0.31	1931	MS, RI
15	Isoamyl benzoate	94-46-2	1944.08 ± 1.33	1937	MS, RI
16	Phenylpropyl acetate	122-72-5	1957.25 ± 0.48	1965	MS, RI
17	Creosol	93-51-6	1957.36 ± 0.89	1952	MS, RI
18	Pentyl benzoate	2049-96-9	2004.46 ± 0.22	2017	MS, RI
19	Methyleugenol	93-15-2	2032.82 ± 0.54	2030	MS, RI, Std
20	(E)-Cinnamaldehyde	14371-10-9	2060.17 ± 0.59	2063	MS, RI, Std
21	3-Phenyl-1-propanol	122-97-4	2060.17 ± 0.59	2049	MS, RI
22	(E)-Methyl cinnamate	1754-62-7	2099.75 ± 0.69	2096	MS, RI
23	Cinnamyl formate *	104-65-4	2122.36 ± 0.58	2094	MS
24	Hexyl benzoate *	6789-88-4	2122.36 ± 0.58	2096	MS
25	3-Buten-2-one, 4-phenyl- *	1896-62-4	2137.34 ± 1.32	2103	MS
26	3-Hexen-1-ol, benzoate, (Z)-	25152-85-6	2166.68 ± 0.67	2148	MS, RI
27	(E)-Cinnamyl acetate	21040-45-9	2175.61 ± 0.89	2182	MS, RI, Std
28	Eugenol	97-53-0	2180.34 ± 1.15	2186	MS, RI, Std
29	(E)-Methyl isoeugenol	6379-72-2	2207.29 ± 0.40	2209	MS, RI, Std
30	cis-Isoeugenol	5912-86-7	2285.01 ± 0.74	2288	MS, RI
31	(E)-Cinnamic alcohol	4407-36-7	2312.04 ± 1.41	2294	MS, RI, Std
32	4-allylphenol	501-92-8	2371.78 ± 1.24	2358	MS, RI, Std
33	trans-Isoeugenol	5932-68-3	2394.44 ± 0.99	2383	MS, RI, Std
34	Benzyl ether *	103-50-4	2455.79 ± 1.57	2356	MS
35	5-Indanol *	1470-94-6	2505.96 ± 1.62	-	MS
36	2-Allylphenol *	1745-81-9	2560.28 ± 1.86	2132	MS
37	Benzyl Benzoate	120-51-4	2642.06 ± 1.70	2655	MS, RI, Std
	**Fatty acid derivatives**				
38	Acetic acid, methyl ester	79-20-9	867.67 ± 0.51	856	MS, RI
39	Hexanal	66-25-1	1092.36 ± 1.21	1093	MS, RI
40	2-Hexenal, (E)-	6728-26-3	1226.28 ± 0.32	1232	MS, RI, Std
41	1-Hexanol	111-27-3	1358.07 ± 0.64	1360	MS, RI
42	3-Hexen-1-ol *	544-12-7	1387.03 ± 0.43	1388	MS, RI
43	2-Hexen-1-ol(E)	928-95-0	1409.33 ± 0.39	1415	MS, RI
44	Nonanal	124-19-6	1424.66 ± 0.50	1406	MS, RI
45	Acetic acid	64-19-7	1459.71 ± 1.78	1461	MS, RI
46	6-Hepten-1-ol, 2-methyl- *	67133-86-2	1478.84 ± 0.66	1467	MS, RI
47	2-Nonenal *	2463-53-8	1535.32 ± 0.85	1546	MS, RI
48	1-Nonanol	143-08-8	1683.66 ± 1.47	1676	MS, RI
49	2,4-Decadienal *	2363-88-4	1852.02 ± 0.73	1811	MS
50	Alkane1		2036.25 ± 0.69		MS
51	Nonanoic acid	112-05-0	2197.06 ± 0.61	2194	MS, RI
52	Alkane2		2145.80 ± 1.24		MS
53	Alkane3		2265.75 ± 0.74		MS
54	Alkane4		2520.36 ± 1.35		MS
	**Terpenoids**				
55	Fenchol *	1632-73-1	1641.38 ± 1.43	1591	MS
56	Bornyl acetate *	76-49-3	1643.92 ± 0.91	1597	MS
57	(E)-α-Elemene *	5951-67-7	1675.75 ± 0.63	1695	MS, RI
58	Borneol *	507-70-0	1721.42 ± 0.52	1719	MS, RI
59	Carvone *	99-49-0	1769.73 ± 0.86	1751	MS, RI
60	(Z,E)-α-Farnesene *	26560-14-5	1793.17 ± 1.05	1737	MS
61	α-Farnesene *	502-61-4	1764.04 ± 0.99	1758	MS, RI
62	Carveol *	1197-07-5	1858.35 ± 1.28	1845	MS, RI
63	trans-β-Ionone	79-77-6	1971.82 ± 1.16	1958	MS, RI
64	Dihydro-β-ionol	3293-47-8	1991.87 ± 0.74	1977	MS, RI
	**others**				
65	Dimethyl sulfide	75-18-3	-	753	MS

“*” indicates that the compound may be other isomers. “-” representative not found or not measured. MS, RI, and Std represent mass spectrometry comparison (match factor > 75), retention index comparison (deviation < 20), and standard comparison, respectively. “**” Measured RI was calculated using the RTs of 9 QC sample data and expressed as “mean ± SD”.

**Table 2 plants-12-00308-t002:** Contents of fifteen differential volatile compounds in six cultivars of *P. mume* blossoms.

Number	Compound Name	CAS-Number	VIP	Contents μg/g					
				YD	JM	GLH	LE	DFZS	GF
1	Benzaldehyde	100-52-7	3.606	157.00 ± 9.21	112.00 ± 4.21	108.02 ± 6.00	126.76 ± 9.83	62.61 ± 7.60	149.91 ± 8.06
2	Methyl benzoate	93-58-3	3.085	172.36 ± 7.23	207.26 ± 6.85	166.90 ± 5.90	210.76 ± 11.67	212.20 ± 6.35	157.84 ± 8.82
3	Ethyl benzoate	93-89-0	1.002	0.28 ± 0.11	1.13 ± 0.38	2.38 ± 0.49	0.54 ± 0.27	0.46 ± 0.14	1.21 ± 0.79
4	Benzyl acetate	140-11-4	1.762	15.99 ± 1.21	8.69 ± 2.15	4.34 ± 0.38	11.29 ± 1.62	0.84 ± 0.15	5.80 ± 1.16
5	Benzyl alcohol	100-51-6	2.202	41.41 ± 5.73	13.81 ± 2.66	19.68 ± 3.19	23.67 ± 3.49	6.82 ± 1.33	22.65 ± 2.73
6	Methyleugenol	93-15-2	1.392	5.11 ± 0.57	1.53 ± 0.38	4.83 ± 0.30	1.97 ± 0.67	2.60 ± 0.58	2.06 ± 0.33
7	(E)-Cinnamaldehyde	14371-10-9	1.797	0.91 ± 0.30	0.26 ± 0.09	27.52 ± 7.41	0.29 ± 0.16	1.89 ± 0.31	25.09 ± 2.48
8	(E)-Cinnamyl acetate	21040-45-9	2.336	1.28 ± 0.48	1.48 ± 0.84	50.96 ± 8.52	0.34 ± 0.15	0.42 ± 0.08	52.22 ± 3.68
9	Eugenol	97-53-0	2.580	64.87 ± 3.39	43.03 ± 3.47	48.80 ± 3.25	45.79 ± 5.18	56.94 ± 5.06	34.62 ± 4.05
10	(E)-Methyl isoeugenol	6379-72-2	1.244	0.73 ± 0.16	1.51 ± 0.27	0.61 ± 0.37	0.56 ± 0.17	0.75 ± 0.11	0.49 ± 0.10
11	Alkane3		1.442	14.10 ± 2.72	4.92 ± 0.86	6.30 ± 1.00	9.13 ± 2.75	8.86 ± 2.65	3.97 ± 0.99
12	(E)-Cinnamic alcohol	4407-36-7	2.107	0.44 ± 0.08	0.18 ± 0.09	36.33 ± 6.25	0.11 ± 0.08	0.30 ± 0.07	40.00 ± 7.30
13	4-allylphenol	501-92-8	3.265	40.04 ± 9.10	15.53 ± 1.98	7.52 ± 2.89	9.88 ± 3.79	34.19 ± 5.29	10.92 ± 2.08
14	unknown 19		1.617	7.84 ± 1.82	14.20 ± 1.23	5.61 ± 1.13	11.89 ± 1.97	10.22 ± 3.71	6.32 ± 0.92
15	Benzyl Benzoate	120-51-4	2.749	63.29 ± 11.21	40.94 ± 4.13	37.39 ± 5.58	54.74 ± 9.51	64.67 ± 16.78	35.52 ± 9.53

VIP, variable importance of projection value.

**Table 3 plants-12-00308-t003:** Aroma description of nine volatile metabolites VIP > 1 based on their OAVs in six cultivars of *P. mume* blossoms.

No.	Compound Name	Olfactory Threshold/μg/g	VIP	OAV s					
				YD	JM	GLH	LE	DFZS	GF
1	Benzaldehyde	0.2 ^a^	1.323	785.00 ± 46.04	560.00 ± 21.04	540.10 ± 30.02	633.82 ± 49.13	313.06 ± 37.98	749.56 ± 40.27
2	2-Nonenal *	0.0005 ^a^	1.054	132.79 ± 112.21	-	198.28 ± 97.27	-	-	65.09 ± 78.30
3	Methyl benzoate	0.00052 ^a^	2.804	331,469.15 ± 13,902.30	398,572.02 ± 13,170.47	320,964.28 ± 11,343.53	405,298.09 ± 22,440.74	408,084.40 ± 12,208.62	303,542.79 ± 16,951.69
4	Benzyl acetate	0.002 ^a^	2.578	7993.82 ± 604.97	4345.63 ± 1073.17	2168.16 ± 191.01	5643.68 ± 812.39	418.15 ± 74.81	2901.00 ± 579.78
5	3,4-Dimethoxytoluene *	0.053 ^a^	1.198	55.44 ± 5.31	15.88 ± 7.43	26.54 ± 9.28	102.29 ± 13.83	25.89 ± 9.85	37.98 ± 24.99
6	Trans-β-Ionone	0.000007 ^a^	2.502	64,831.31 ± 16,666.15	58,367.53 ± 4829.57	33,826.24 ± 22,943.26	30,862.43 ± 8876.74	21,560.30 ± 11,394.21	37,296.25 ± 11,507.64
7	Eugenol	0.007 ^a^	2.504	9267.71 ± 483.68	6147.39 ± 496.08	6971.31 ± 464.79	6540.78 ± 739.34	8133.62 ± 722.27	4946.27 ± 578.28
8	(E)-Cinnamic alcohol	0.077 ^a^	1.244	5.70 ± 1.06	2.34 ± 1.16	471.77 ± 81.23	1.39 ± 1.00	3.89 ± 0.89	519.48 ± 94.88
9	4-allylphenol	0.019 ^b^	2.356	2107.63 ± 479.13	817.17 ± 104.23	395.70 ± 151.85	519.90 ± 199.47	1799.25 ± 278.14	574.55 ± 109.20

^a^ Olfactory thresholds were from Compilations of Flavor Threshold Values in Water and Other Media [23]. ^b^ From LH Ma (2021). Ref. [24]. VIP, variable importance of projection value; OAVs, odor activity values. “*” indicates that the compound may be other isomers.

**Table 4 plants-12-00308-t004:** Comparisons of the differential compounds screened by the contents of 89 detected and OAVs of 65 identified compounds in different cultivars.

No.	Compound Name	Olfactory Threshold/μg/g	VIP by		Aroma Descriptions ^c^	GLH		GF		YD		DFZS		JM		LE	
			Contents	OAVs		C	O	C	O	C	O	C	O	C	O	C	O
1	Benzaldehyde	0.2 ^a^	3.606	1.323	Sharp, sweet, bitter, almond, cherry												
2	2-Nonenal *	0.0005 ^a^		1.054	Fatty, green, waxy, cucumber, melon												
3	Methyl benzoate	0.00052 ^a^	3.085	2.804	Phenolic, wintergreen, almond, floral, cananga odorata												
4	Ethyl benzoate	0.053 ^a^	1.002		Fruit, musty, sweet, wintergreen												
5	Benzyl acetate	0.002 ^a^	1.762	2.578	Sweet, floral, fruit, Jasmine, fresh												
6	3,4-Dimethoxytoluene *	0.053 ^a^		1.198	Fruit, musty, sweet, wintergreen												
7	Benzyl alcohol	5.5 ^a^	2.202		Floral, rose, phenolic, spice												
8	Trans-β-Ionone	0.000007 ^a^		2.502	Seaweed, violet, flower, raspberry												
9	Methyleugenol	0.068 ^a^	1.392		Sweet, fresh, warm, spicy, clove, cinnamon												
10	(E)-Cinnamaldehyde	6 ^a^	1.797		Sweet, spicy, candy, cinnamon, warm												
11	(E)-Cinnamyl acetate	0.15 ^a^	2.336		Sweet, floral, spicy, spice												
12	Eugenol	0.007 ^a^	2.580	2.504	Sweet, spicy, clove, woody												
13	(E)-Methyl isoeugenol	n.a.	1.244		n.a.												
14	Alkane3	n.a.	1.442		n.a.												
15	(E)-Cinnamic alcohol	0.077 ^a^	2.107	1.244	Sweet, spice, hyacinth, spicy, cinnamon												
16	4-allylphenol	0.019 ^b^	3.265	2.356	Phenolic, medicinal, herbal												
17	unknown 19	n.a.	1.617		n.a.												
18	Benzyl Benzoate	0.341 ^a^	2.749		Sweet, spice, floral, fruit												

“*” indicates that the substance may also be other isomers of the compound. “n.a.” OAV not available. C, content; O, OAV. ^a^ Olfactory thresholds were from Compilations of Flavor Threshold Values in Water and Other Media [23]. ^b^ From LH Ma (2021) [24]. ^c^ Aroma Descriptions were from the database (www.perflavory.com) (accessed on 6 April 2022) [26]. VIP, variable importance of projection value; OAVs, odor activity values.

## Data Availability

Not applicable.

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
