# Peer review of "Volatile Metabolome and Aroma Differences of Six Cultivars of Prunus mume Blossoms"

_plants, 2023, doi:10.3390/plants12020308_

Round 1
Reviewer 1 Report
The manuscript titled "Volatile metabolome and aroma differences of six cultivars of Prunus mume blossoms" studies a traditional Chinese plant with ornamental values, which has a unique fragrance. The analyzes of this study are focused on the volatile metabolome and aroma of Prunus mume flowers. Six plant varieties are studied, and the volatile metabolites, 89 identified, were analyzed by the HS-SPME-GC-MS technique.
The manuscript is an interesting one, the authors have worked hard. It is well organized and easy to follow.
The abstract provides relevant information on the entire manuscript.
The introduction is comprehensive and provides information from relevant literature.
Results and discussion are detailed, but I have a small suggestion:
Line 105 In text P. mume is written in italics. Please check all the text
In Table 1 - in the title P. mume is written in italics
In Table 1 - in the footnotes, the abbreviations should be detailed: MS, RI, Std
Statistical analysis is missing from table 1
In Table 2 - I recommend footnotes for abbreviations
In Table 2, I don't think 3 decimal places are necessary for the standard deviation
In Table 3 I recommend footnotes for abbreviations
Reviewer 2 Report
Manuscript ID: plants-2120765
Type of manuscript: Article
Title: Volatile metabolome and aroma differences of six cultivars of Prunus
mume blossoms
Authors: Ting Li, Xi Zhao, Xueli Cao *
Submitted to section: Phytochemistry
Comments to the manuscript
The manuscript entitled “Volatile metabolome and aroma differences of six cultivars of Prunus mume blossoms” is very well structured and it presents a comprehensive investigation on the aroma characteristics of P. mume blossoms. The ideas are clearly described and the methodology is adequately chosen.
Indeed, the aroma of P. mume blossoms has attracted a great deal of attention so that an article on the headspace volatiles, extracts and aroma of eight P. mume cultivars recently has been published in Molecules journal. Undoubtedly, the authors are familiar with the mentioned article as it is referred and discussed in the introduction.
In the Abstract
The research is well summarized; the methodology and findings are clearly described.
1- For better understanding it is advisable to introduce the cultivars in the beginning:
Row 11- six extensively planted cultivars of P. mume blossoms (GLH, GF, JM, LV, YD and DFZS)
2- In the whole manuscript Prunus mume or P. mume must be written in italic.
In the Introduction
The importance of the investigation is very well explained and the existing research has been introduced.
3- Row 64- define more accurately: 74 compounds in endogenous extracts
In the Results and discussion
The results are explained in a logical sequence and supported with figures and tables. The identification of compounds was performed accurately with the names written after JUPAC nomenclature. It is essential to mention that the component quantification was done by using internal standard to give the real amount (μg/g) in the sample. On this way it is easier to see the quality features of the different samples as by using the relative content (%).
Two of the cultivars (JM and DFZS) were studied previously by the same technique HS-SPME-GC (Molecules. 2021, 26, 7256) and there are differences in the presence and quantity of the main components especially in the content of methyl benzoate, benzyl alcohol, eugenol and 4-allylphenol, and not only. Why the authors did not discussed these facts?
The authors gave only one explanation concerning the regulation of expression of enzymes catalyzing the relevant synthetic reactions.
The differences in the contents may also be the results of using different analytical methods. In this case for example were used different GC columns. Also the identification methodology is a crucial factor.
Nevertheless, I find their results relevant.
In the Materials and Methods
This section is described clearly so the methodology could be repeated. The analytical technique HS-SPME-GC-MS combined with chemometrics is well known tool in the metabolomics, with the special feature of this research, it is excellent performed.
